# Effect of *Lactobacillus reuteri* S5 Intervention on Intestinal Microbiota Composition of Chickens Challenged with *Salmonella enteritidis*

**DOI:** 10.3390/ani12192528

**Published:** 2022-09-22

**Authors:** Shuiqin Shi, Duoqi Zhou, Yuan Xu, Jinsheng Dong, Yu Han, Guangyu He, Wenhui Li, Jie Hu, Yannan Liu, Kai Zhao

**Affiliations:** School of Life Sciences, Anhui Key Laboratory of Biodiversity Research and Ecological Protection in Southwest Anhui, Anqing Normal University, Anqing 246133, China

**Keywords:** *L. reuteri* S5, intestinal microbiota, infection, *S. enteritidis*

## Abstract

**Simple Summary:**

The interaction between intestinal microbiota and host plays a key role in the development of intestinal diseases. *Salmonella enteritidis* infection is a common cause of human gastroenteritis, which can lead to gastrointestinal dysfunction and microecological disorder. It is an important public health problem all over the world. Lactic acid bacteria are among the main probiotics in the intestinal microbiota, which is one of the main defense lines against intestinal pathogens. It is now also considered to be an effective alternative antibiotic to fight pathogenic bacterial infections. In order to explore the potential mechanism of lactic acid bacteria against *S. enteritidis* infection, this study started from the perspective of lactic acid bacteria regulating the intestinal microbiota and the infection of antigenic bacteria. The probiotic strain *Lactobacillus*
*reuteri* S5 (*L. reuteri* S5) was used to establish an animal model of *S. enteritidis* infection in broilers. The regulatory effect of *L.*
*r**euteri* S5 on the intestinal microbiota structure of chickens infected with *S. enteritidis* is studied by high throughput sequencing technology. The results showed that *L. reuteri* S5 could regulate the composition and abundance of intestinal microbiota and resist the infection of *S. enteritidis*.

**Abstract:**

To understand the mechanism of lactic acid bacteria against *Salmonella enteritidis* infection; we examined how lactic acid bacteria regulated the intestinal microbiota to resist infection by pathogenic bacteria. The probiotic strain *Lactobacillus reuteri* S5 was used to construct an animal model of *S. enteritidis* infected broilers. A high-throughput sequencing technology was used to analyze the regulatory effects of *L. reuteri* S5 on the structure of the intestinal microbiota of broilers infected with *S. enteritidis*; and to examine the possible defense mechanism they used. Our results showed that the administration of *L. reuteri* S5 reduced colonization of *S. enteritidis* (*p* < 0.05), decreased intestinal permeability (*p* < 0.05), and reduced the bacterial displacement likely due by *S. enteritidis* colonization (*p* < 0.05), suggesting some enhancement of the intestinal barrier function. Furthermore, *L. reuteri* S5 increased the number of operational taxonomic units (OTUs) in the chicken cecal microflora and the relative abundance of Lactobacillaceae and decreased the relative abundance of Enterobacteriaceae. These results suggest that the lactic acid bacterium *L. reuteri* S5 protected the intestinal microbiota of chickens against *S. enteritidis* infection.

## 1. Introduction

Salmonella is a common cause of human gastroenteritis. *Salmonella* subsp. *Salmonella enteritidis* is an important foodborne pathogen that causes human salmonellosis and can be traced back to poultry and poultry products [1]. Epidemiological studies in various countries have shown that poultry-derived food is one of the commonest vectors of *Salmonella* [2]. Salmonellosis is an important public health problem throughout the world, and studies have shown that *S. enteritidis* infection can affect the formation of the intestinal microbiota in chicks [3]. The intestinal microbiota and intestinal homeostasis play key roles in the control of *Salmonella* transmission, infection, and disease [4,5].

Gastrointestinal infection is an important disease that indirectly damages the chicken industry [6]. Regulating the intestinal microflora of chickens by directly feeding them beneficial microorganisms and probiotic agents can reduce this challenge to intestinal health [7]. Probiotics are among the most popular bioactive and health-promoting food additives [8]. It is noteworthy that probiotics are sometimes referred to as “direct feed microorganisms” (DFM) [9] when used for livestock, and have been shown to improve growth performance to levels similar to those achieved with antibiotic growth promoters (AGPs). Because AGPs are banned or restricted in many countries, the poultry industry faces an increasing challenge in dealing with intestinal pathogens such as *Salmonella*. Therefore, it is very important to maintaining the stability of the internal environment of the intestinal microbiota and prevent the infection of pathogenic bacteria in the early stage of poultry growth to promote healthy breeding of poultry.

Lactic acid bacteria are the bacterial strains most commonly used as probiotics. After *S. heidelberg* (SH)-infected turkeys were treated with lactic acid bacteria, the colonization of the turkey cecum by SH was eliminated. Lactic acid bacteria can improve the safety of poultry food products by reducing the colonization of poultry by foodborne human pathogens, including *Salmonella* [10] and *Campylobacter* [11]. Studies have shown that the composition and diversity of the intestinal flora are very important to resist colonization by intestinal pathogens, especially in the early stages of animal intestinal development when the microecosystem has not been fully established [12]. Thus, members of the genus *Bacteroides* produce short-chain fatty acids and propionate to mediate resistance to *S. enterica serovar Typhimurium* infection, and indirectly restrict pathogen growth by disturbing the acid base balance of the intestinal environment [13,14]. Moreover, the intestinal microbiota can produce indole to reduce the expression of the virulence genes of *S.*
*Typhimurium* and inhibit its colonization of the host [15]. In this context, *Lactobacillus reuteri* is the microorganism that occurs most frequently in the intestines and is present in the intestines of almost all vertebrates [16]. It is currently widely used as a probiotic, and has been shown to inhibit *Streptococcus mutans* growth [17,18], regulate the composition of human intestinal microbiota [19], reduce the incidence rate of acute diarrhea in children [20], and maintain the intestinal barrier integrity and immune system regulation [21]. It is an important microorganism used widely in foods, drugs, and livestock and poultry feed additives.

To investigate the mechanism of action of *L. reuteri* against *S. enteritidis* infection, we examined how *L. reuteri* regulated the intestinal barrier and prevented infection by pathogenic bacteria. To evaluate the anti-infection activity of *L. reuteri* S5, we constructed an animal model of *S. enteritidis* infection in broiler chickens and analyzed the regulatory effects of *L. reuteri* S5 on the structure of the intestinal microbiota of infected chickens. We investigated the mechanism by which lactic acid bacteria regulated the intestinal microbiota of broiler chickens to resist *S. enteritis* infection, in order to lay the foundation for a strategy to use *L. reuteri* S5 as a prophylactic against *S. enteritidis* infection in chickens.

## 2. Materials and Methods

### 2.1. Experimental Animals

In total, broilers (1-day-old chicks, n = 180) were obtained from a local parent stock supplier and randomly transferred to compact-type four-tier cages, five chicks per cage. Battery cages were equipped with wire mesh, dropping trays, nipple drinkers, and trough feeders. The battery cages were placed in an environmentally controlled room with windows. The experiment consisted of four dietary treatments and was set up in a completely randomized design in which 45 chicks were randomly divided into four experimental groups (control group (C), *L. reuteri* S5 group (L), *S. enteritidis* infection group (S), and *L. reuteri* S5 + *S. enteritidis* group (LS)). Throughout the experimental cycle, the C and S groups were fed a basic diet and given 1 mL of sterile phosphate-buffered saline (PBS) every day by gavage. The chickens in the S group were administered a suspension of *S. enteritidis* (1.0 × 10^9^ colony-forming units [CFU]/mL) daily until the age of 14 days. The L and LS groups were administered a suspension of *L. reuteri* S5 (1.5 × 10^8^ CFU/mL) every day for the whole experimental cycle, until the age of 14 days, and the chickens in the LS were also administered a suspension of *S. enteritidis* (1.0 × 10^9^ CFU/mL). The experiment lasted for 14 days, and the chicks were fed the experimental diets throughout the experimental period. Chicks had free access to feed and water.

### 2.2. Determination of pH in Chicken Intestinal Tract

The abdominal cavity of the chicken was cut, and the cecum was taken out. The pH value of chicken cecal contents was immediately measured with a portable pH (PHS-3C, Shanghai Bioengineering Co., Ltd. Shanghai, China) meter and repeated three times in each group.

### 2.3. Determination of Intestinal Permeability

Intestinal permeability was assessed with a fluorescein isothiocyanate (FITC)-dextran assay. Three chickens in each group were randomly selected and fasted for 4 h. Then, 100 μL of FITC–dextran solution was tube feeding at the dose of 60 μg/100 g bodyweight. After 4 h, the chick heart blood was collected and placed in an Eppendorf tube at room temperature in dark. The blood was then centrifuged at 2000× *g* for 15 min, and the serum was collected and stored in the dark. A serum sample (30 μL) was diluted 1:1 with PBS and added to a 96-well black plate. The fluorescence of each serum sample was measured spectrophotometrically at an excitation wavelength of 485 nm and an emission wavelength of 535 nm.

### 2.4. Bacterial Translocation

About 1.0 g of heart, liver, spleen, lung, and kidney tissues of the chickens were quickly isolated under sterile conditions, rinsed with precooled sterile PBS, dried with disposable sterile filter paper, placed in 2 mL sterile EP (Eppendorf tubes), and weights were recorded. Triton X-100 (1%, 1 mL) was added to each sample, which was then ground in a high-throughput tissue grinder. The samples were then placed on ice for 15 min to allow complete lysis, and serially diluted 10-fold to the appropriate concentrations. Then, 10 μL were pipetted onto the prepared *Salmonella* selective medium SS Agar solid medium, and placed in a 37 °C incubator for 24 h. The colonies on each plate were then counted. Three repetitions of each experimental sample were processed. Finally, the bacterial content per gram of tissue (CFU/g) was calculated based on the weight of the tissue and the number of bacteria.

### 2.5. Determination of Total Lactic Acid Bacteria and Salmonella in Cecal Feces

After each chicken was bled from the neck and killed, the cecum was excised and ligated at both ends with a sterile cord, and quickly returned to the laboratory at low temperature. A sample of the cecal intestinal contents (0.5 g) was collected aseptically in a sterile ultraclean platform and placed in a sterile 10 mL centrifuge tube. Sterile normal saline (4.5 mL) was added to each tube, and the samples were mixed on a vortex shaker and diluted 10^6^–10^9^ fold. Each experimental group was set up with 3 repetitions, and the prepared mixture was sucked with a pipette gun for 10 μL, respectively, and then dropped on the lactic acid bacteria medium MRS agar medium and SS medium to dry. The medium was placed in a 37 °C incubator for 24 h, after which the colonies were counted. The bacterial counts were calculated with the plate counting method and expressed as the logarithm of the number of bacteria cultured in each gram of chicken cecal contents (CFU/g).

### 2.6. High-Throughput DNA Sequencing Analysis of Chicken Cecal Microbiota

Microbial DNA was extracted with the QIAamp Fast DNA Stool Mini Kit (QIAGEN, Frankfurt, Germany). The final DNA concentration and purity were determined with a NanoDrop 2000 UV-Vis spectrophotometer (Thermo Fisher Scientific, Waltham, MA, USA). An optical density ratio (OD_260/280_) of 1.8–2.0 indicated optimal DNA purity. The V3–V4 hypervariable region of the bacterial 16S rRNA gene was amplified with primers 338F (5′-ACTCCTACGGGAGGCAGCAG-3′) and 806R (5′-GGACTACHVGGGTWTCTAAT-3′) on a PCR thermocycler system, and the products were purified with an AxyPrep DNA Gel Extraction Kit (Axygen Biosciences, Union City, CA, USA), and quantified with a QuantiFluor™-ST System fluorometer (Promega, Madison, WI, USA). Equimolar amounts of the purified amplicons were pooled and paired-end sequenced on the Illumina MiSeq platform, with standard protocols.

### 2.7. Statistical Analysis

The raw FASTQ files were demultiplexed and quality filtered with Trimmomatic, and then merged with FLASH. Sequences with ≥ 97% similarity were clustered as operational taxonomic units (OTUs) with UPARSE (version 7.1, Shanghai Meiji Biomedical Technology Co., Ltd., Shanghai, China), and chimeric sequences were identified and removed with UCHIME. The taxonomy of each 16S rRNA gene sequence was analyzed with the RDP Classifier algorithm against the Silva 16S rRNA database, using a confidence threshold of 70%. These sequences were clustered into OTUs (97% similarity) with UCLUST. The reference OTU sequences were taxonomically assigned with the UCLUST Consensus Taxonomy Assigner against the Greengenes Database, with a 0.5 confidence threshold, and identified to the species level. The unweighted UniFrac distances were used to compare the bacterial communities depending on the chicken. Based on the sample information, a redundancy analysis with clustered OTUs was used to compare the chicken with the bacterial community structures using the R statistical software (version 3.3.0). To determine whether the chicken specific microbiomes were significantly distinguishable, we used analysis of similarities, a nonparametric statistical test. The significance of differences between the groups was determined with permutations, using the vegan package in the R statistical software (Shanghai Meiji Biomedical Technology Co., Ltd., Shanghai, China).

## 3. Results

### 3.1. Effect of L. reuteri S5 on Intestinal Barrier Integrity

As shown in Figure 1A, the pH of the chicken cecal contents tended to be lower in group L than in groups S and C, but the differences were not significant (*p* > 0.05). Intestinal permeability was different in each group, as shown in Figure 1B. Intestinal permeability tended to be slightly lower in group L than in group C, but the difference was not significant (*p* > 0.05). Intestinal permeability was significantly higher in group S than in group C (*p* < 0.0001) and was significantly lower in group LS than in group S (*p* < 0.0001).

To evaluate the protective effect of *L. reuteri* S5 on the chicken intestine, the *Salmonella* bacteria present in the ceca were counted in groups LS and S. The bacterial load of liver, lung and kidney in group LS was significantly lower than that in group S (*p* < 0.05). There was no significant difference in the bacterial load of heart and spleen between the two groups (*p* > 0.05), as shown in Figure 2.

### 3.2. Effects of L. reuteri S5 on Intestinal Microbiota of Chicken Ceca

The colony counts of lactic acid bacteria in the chicken ceca were significantly higher in group L than in group C (*p* < 0.05) and were higher in group LS than in group S, but the difference was not significant (*p* > 0.05), as shown in Figure 3A. The colony counts of *Salmonella* in the chicken ceca were significantly lower in group L than in group S (*p* < 0.01), and the counts of *Salmonella* were also lower in group LS than in group S, but the difference was not significant (*p* > 0.05), as shown in Figure 3B.

The number of OTUs in each group of samples is shown in Figure 4A. The average number of OTUs was 304 in the group C samples; 352 in the group L samples; 291 in the group S samples; and 343 in the LS group samples. The species richness was significantly higher in group L than in group C, whereas it was lower in group S than in group C. In this study, the species diversity and number of annotated OTUs were lower in group S than in group C or group L. When *L. reuteri* S5 was fed to chicks from 1 day of age, the microbial species diversity in the chicken ceca was higher in group L than in group C, and the number of annotated OTUs was significantly higher in group L than in group C (*p* < 0.05).

The structure of the chicken intestinal microbiota at the family level was analyzed in each group, as shown in Figure 4B. The data are annotated to 10 families in total. The abundance of the family Enterobacteriaceae was significantly lower in group L (19.6%) than in group C (48.0%, *p* < 0.01), but was significantly higher in group S (47.9%) than in group LS (24.7%, *p* < 0.05). The abundance of Lactobacillaceae was significantly higher in group L (15.3%) than in group C (6.5%, *p* < 0.01), whereas its abundance was significantly lower in group S (1.3%) than in group LS (7.4%, *p* < 0.001). The abundance of Clostridiaceae was higher in group L than in the other groups. Ruminococcaceae accounted for 23.0%, 34.6%, 32.3%, and 32.0% in groups C, L, LS, and S, respectively.

The cluster graph shown in Figure 5 compares the species richness in the ceca of the four groups of chickens. At the genus level, the relative abundance of *Lactobacillus* was highest in the ceca of group L (11.00%), where it was significantly higher than in group C (*p* < 0.01), and also higher than that in group S (*p* < 0.01). The relative abundance of *Lactobacillus* was lowest in group S (1.39%, *p* < 0.01). On the contrary, the relative abundance of *Escherichia–Shigella* was highest in the chicken ceca of group S (49.50%), where it was significantly higher than in group LS (12.95%, *p* < 0.01); its relative abundance was lowest in group L (3.92%, *p* < 0.01). The species composition of group LS was close to that of group L.

### 3.3. LEfSe Analysis of the Effect of L. reuteri S5 on the Abundance of Chicken Intestinal Microbiota

An LEfSe analysis was used to compare the bacterial abundances in the chicken intestinal microbiota. The results are shown in Figure 6. At the family level, the L group abundance of species in the family Ruminococcaceae, Bacillaceae, Mollicutes_RF39, Caulobacteraceae, and Defluviitaleaceae were significantly more abundant than in group C. The species abundance of Enterobacteriaceae was significantly higher in group C than in group L.

As shown in Figure 7, the abundances of species in the families Lachnospiraceae and Lactobacillaceae were significantly higher in group LS than in group S, whereas those of Peptostreptococcaceae and Clostridiaceae_1 were significantly higher in group S than in group LS.

## 4. Discussion

*Salmonella* is one of the most widespread foodborne disease organism [22], and can be transmitted to humans through animal-based foods [23]. In this study, we found that the level of serum FITC-dextran increased significantly after chickens were infected with *S. enteritidis*. FITC-dextran does not penetrate the intestinal barrier well, but once the intestinal barrier is damaged and the tight junctions between the epithelial cells are compromised, FITC-dextran molecules can penetrate the intestinal barrier and enter the blood [24]. *Salmonella* infection destroys the tight junctions of the host’s intestinal barrier, resulting in a loss of intestinal barrier function, bacterial translocation, and the penetration of the intestinal barrier by FITC-dextran molecules [25]. In this study, the bacterial load of liver, lung, and kidney in group LS was significantly lower than that in group S (*p* < 0.05). Panpetch studies showed that *Lactobacillus rhamnosus* L34 attenuated the dysbiosis of intestinal flora and bacteremia caused by intestinal leakage in a mouse model of *Candida albicans* induced colitis. [26]. *L. reuteri* S5 strengthened the tight junctions of the chicken intestinal barrier and the integrity of the intestinal epithelial cells layer. These results are consistent with another study that demonstrated that probiotics prevent the translocation of *Salmonella*, inhibit oxidant-induced intestinal permeability, and improve the intestinal barrier function [26]. Therefore, the proper protection and reinforcement of the intestinal barrier function were essential in maintaining the health of the host.

The intestinal flora is closely related to the health of the host. It plays an important role in the intestinal tissue, immunity, energy metabolism and nutrient digestion and absorption of animals. In this study, high-throughput sequencing was used to explore the effects of *L. reuteri* S5 on the composition of intestinal flora of chicks and the regulation of intestinal flora infected by *S. enteritidis*.

In this study, the species diversity and number of annotated OTUs were lower in group S than in group C or group L. When *L. reuteri* S5 was fed to chicks from 1 day of age, the microbial species diversity in the chicken ceca was higher in group L than in group C, and the number of annotated OTUs was significantly higher in group L than in group C (*p* < 0.05). Zhang et al. investigated the effect of *L. reuteri* ZLR003 on the composition of the intestinal microbiota in weaned piglets and found that it increased relative to that in untreated weaned piglets [27]. Other research showed that the diversity of the intestinal microbiota in piglets fed *L. reuteri* I5007 was higher in the cecum [28]. Studies have reported that the species diversity of the intestinal microbiota of animals was high, which benefited the health and productive performance of the host [29].

The cluster graph shown in Figure 5 compares the species richness in the ceca of the four groups of chickens. The species composition of group LS was close to that of group L. The species composition of each group was basically unchanged, but there were great differences in the relative richness of different species, according to the consumption of *L. reuteri* S5 and infection with *S. enteritidis*. The species abundance analysis of group L found that feeding *L. reuteri* S5 increased the abundance of *Lactobacillaceae* and *Clostridiaceae* in the chicken cecum and reduced the species abundance of Enterobacteriaceae. Therefore, we speculated that probiotics affected the total abundance of some microbes in the host intestine but did not alter the overall structure of the intestinal bacterial community in the intestine. Enterobacteriaceae was significantly more abundant in group S than in the other groups (*p* < 0.05). This indicated that *S. enteritidis* infection changed the abundance of the chicken intestinal microbiota. Studies have shown that feeding *L. yoelii* HBUAS54408 restores the abundances of different genera of the families *Lachnospiraceae* and *Clostridiales* after *Salmonella* infection in chickens, and improves the relative abundances of families *Erysipelotrichaceae*, *Christensenellaceae*, and *Peptostreptococaceae* [30]. This is basically consistent with the results of our study.

Interestingly, we found that the relative abundance of *Lactobacillaceae* in chicken caecum in LS group was significantly higher than that in S group (*p* < 0.05), and the relative abundance of *Lachnospiraceae* was also significantly higher than that in other groups. This indicated that *L. reuteri* S5 regulated the relative abundance of *Lactobacillaceae* species in the intestinal microbiota of chickens infected with *S.* Enteritis and improved the levels of some species of the intestinal symbiotic microbiota to jointly resist the damage to the intestinal microbiota structure caused by *S.* Enteritis. We also found a significant negative correlation between *Enterobacteriaceae* and *Lachnospiraceae* in groups C and S, indicating that there may be antagonism between the members of these two taxa, which may affect the overall microbial population in the intestine [31].

## 5. Conclusions

The study showed that the administration of *L. reuteri* S5 reduced colonization of *S. enteritidis*, decreased intestinal permeability, and reduced also the bacterial displacement likely due by *S. enteritidis* colonization, suggesting some enhancement of the intestinal barrier function. The colonization of the intestinal tract by *L. reuteri* S5 increased the number of microbiotal OTUs in the chicken cecum, increased the relative abundance of Lactobacillaceae, and reduced the relative abundance of Enterobacteriaceae. These results suggest that the use of lactic acid bacterium *L. reuteri* S5 in chickens may regulate the intestinal microbiota composition to resist infection by *S.* Enteritis.

## Figures and Tables

**Figure 1 animals-12-02528-f001:**
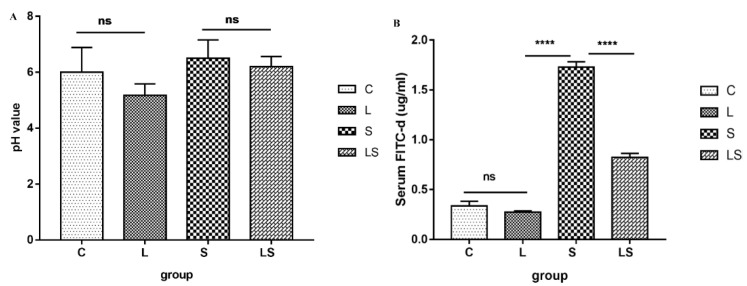
(**A**) pH value of chicken cecum; (**B**) Chicken intestinal permeability. C = control group, L = *L. reuteri* S5 group, S = *S. enteritidis* infection group and LS = *L. reteri* S5 + *S. enteritidis* group. ns indicates that the comparison between the two groups is not significant, **** means *p* < 0.0001.

**Figure 2 animals-12-02528-f002:**
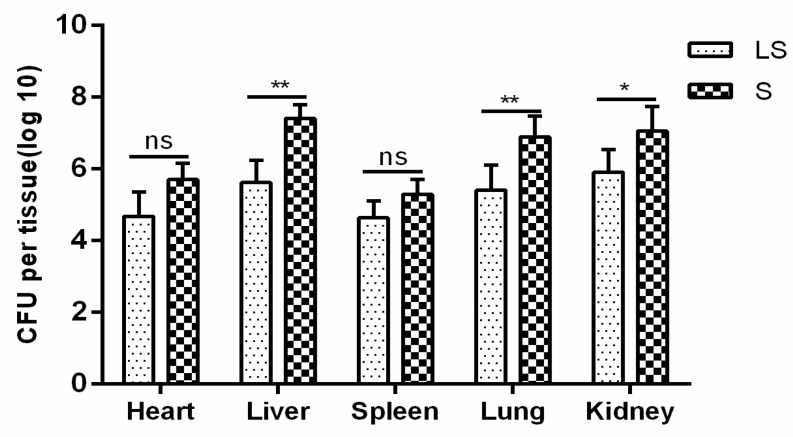
Tissue bacterial count. S = *S. enteritidis* infection group and LS = *L. reteri* S5 + *S. enteritidis* group. ns indicates that the comparison between the two groups is not significant. * means *p* < 0.05, ** means *p* < 0.01.

**Figure 3 animals-12-02528-f003:**
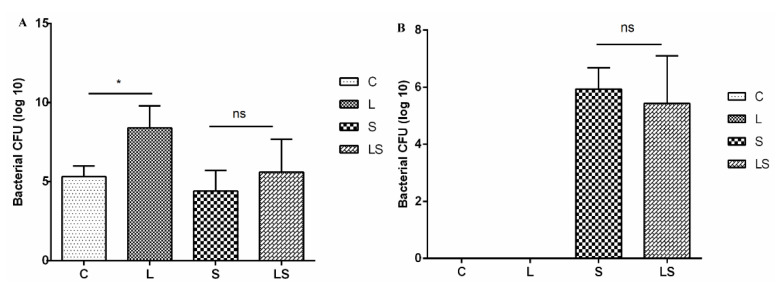
Bacterial count of chicken cecum. (**A**) indicates chicken cecum lactic acid bacteria count, and (**B**) indicates Salmonella count in chicken cecum. C = control group, L = *L. reuteri* S5 group, S = *S. enteritidis* infection group and LS = *L. reteri* S5 + *S. enteritidis* group. ns indicates that the comparison between the two groups is not significant, * means *p* < 0.05.

**Figure 4 animals-12-02528-f004:**
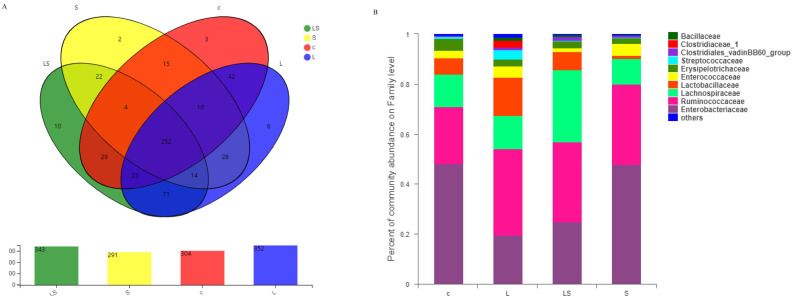
(**A**) shows the number of OTUs of cecal microflora in different groups of chickens; (**B**) represents the relative abundance of species composition in each group at the family level.

**Figure 5 animals-12-02528-f005:**
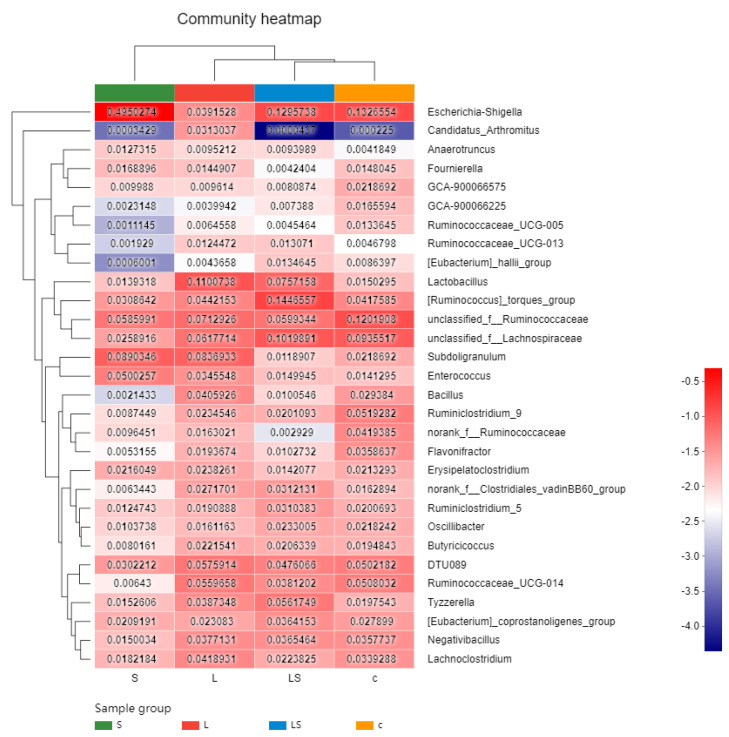
Heat map analysis of relative abundance of cecum intestinal flora of chickens after *L. reuteri* S5 intervention. The variation of the abundance of different species in the sample is displayed through the color gradient of the color block, and the numbers in the figure represent the relative abundance value of the species.

**Figure 6 animals-12-02528-f006:**
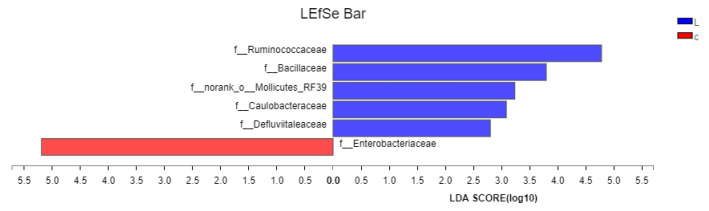
Lefse analysis of intestinal microflora abundance in group L.

**Figure 7 animals-12-02528-f007:**
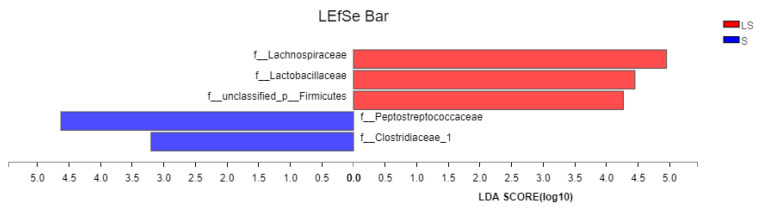
Lefse analysis of intestinal microflora abundance in group LS.

## Data Availability

The data presented in this study are available in the article.

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
