# Peer review of "Effect of Lactobacillus reuteri S5 Intervention on Intestinal Microbiota Composition of Chickens Challenged with Salmonella enteritidis"

_animals, 2022, doi:10.3390/ani12192528_

Round 1
Reviewer 1 Report (Previous Reviewer 1)
The authors have addresses my comments in the resubmitted version and thus I recommend publication
Author Response
Comments and Suggestions for Authors:
The authors have addresses my comments in the resubmitted version and thus I recommend publication.
Response: Thank you very much for your comments. We admire your expertise and patient. Your comments are very useful for our research and publication. Finally, thank you again for your affirmation of the manuscript.
Reviewer 2 Report (Previous Reviewer 2)
see document attached

Author Response
Dear editor and reviewers,
Thank you very much for your review. We admire your expertise and patient. The comments are very useful for our research and publication. We decide to accept all the comments and revise our manuscript carefully according to your comments. The changes were listed blew point by point, and were marked in red in the revised manuscript. The manuscript has greatly benefited from these invaluable suggestions. We invited International Science Editing (http://www.internationalscienceediting.com) for editing this manuscript. We look forward to working with you and the reviewers to move this manuscript closer to publication in the animals.

This manuscript is a resubmission of an earlier submission. The following is a list of the peer review reports and author responses from that submission.
Round 1
Reviewer 1 Report
The hypothesis of the study, i.e. that probiotics may affect the digestive tract bacterial flora, is interesting. However, the results are not nicely presented. The authors should add heat maps or histograms showing the different bacterial taxa identified. The authors mention figures, but for some reason there are no figures within the manuscript. Also I suggest to divide results from discussion. There are a lot of aspects that should be separately discussed, such as the different bacterial taxa, any pathogenic bacterial genera etc.
In my opinion if the research intended to be only mainly focused on probiotic administration effect on Salmonella presence, instead of NGS, a quantitative real time PCR applied on broilers from different groups would be more informative and reliable.
Reviewer 2 Report
See document attached.
